# Transanal Total Mesorectal Excision (TaTME) versus Laparoscopic Total Mesorectal Excision for Lower Rectal Cancer: A Propensity Score-Matched Analysis

**DOI:** 10.3390/cancers14174098

**Published:** 2022-08-24

**Authors:** Yueh-Chen Lin, Ya-Ting Kuo, Jeng-Fu You, Yih-Jong Chern, Yu-Jen Hsu, Yen-Lin Yu, Jy-Ming Chiang, Chien-Yuh Yeh, Pao-Shiu Hsieh, Chun-Kai Liao

**Affiliations:** 1Division of Colon and Rectal Surgery, Department of Surgery, Chang Gung Memorial Hospital, Linkou, No. 5, Fuxing St., Guishan Dist., Taoyuan 33305, Taiwan; 2School of Medicine, Chang Gung University, No. 259, Wenhua 1st Road, Guishan Dist., Taoyuan 33302, Taiwan; 3Division of Colon and Rectal Surgery, Department of Surgery, Chang Gung Memorial Hospital, Keelung Branch, No. 222, Maijin Rd., Anle Dist., Keelung City 20401, Taiwan

**Keywords:** transanal total mesorectal excision (TaTME), rectal cancer, local recurrence, disease free survival, overall survival

## Abstract

**Simple Summary:**

To treat locally advanced rectal cancer with a multimodality approach has led to improved oncological outcomes. Despite the trend of intensification of neoadjuvant therapy, surgery remains the mainstay treatment for rectal cancer. Transanal total mesorectal excision (TaTME) was shown to provide a better distal resection margin, less circumferential resection margin involvement, and a better short-term outcome than laparoscopic TME (LapTME) for mid-low rectal cancer. However, diverse oncological results were reported recently. We aimed to analyze the short- and long-term outcomes of TaTME compared with LapTME in patients with lower rectal cancer. Our results showed that TaTME had similar histopathological results and postoperative outcomes as LapTME, even in the learning curve. However, a better DFS (72% vs. 56.6%, *p* = 0.038) and fewer LR events (9.5% vs. 23.8%, *p* = 0.031) were observed after TaTME. Thus, TaTME can be considered a safe and feasible approach in patients with low rectal cancer.

**Abstract:**

Studies have reported positive short-term and histopathological results of transanal total mesorectal excision (TaTME) for mid-low rectal cancer. The long-term oncological outcomes are diverse, and concerns regarding the high local recurrence (LR) rate of TaTME have recently increased. We retrospectively analyzed 298 consecutive patients who underwent Laparoscopic TME (LapTME) or TaTME between January 2015 and December 2019. Propensity score-matching (PSM) was performed with patients matched for demographics and stage. After PSM, 63 patients were included in each group. The TaTME group had a longer mean operative time (394 vs. 333 min, *p* < 0.001). The blood loss, diverting stoma rate, and conversion rate were similar. Postoperatively, TaTME and LapTME had compatible complications, recovery, and hospital stay. A similar specimen quality was detected in both groups. After a mean follow-up period of 41–47 months, TaTME had less LR than LapTME (9.5% vs. 23.8%, *p* = 0.031). The 3-year overall survival was 80.3% in the TaTME group and 73.6% in the LapTME group (*p* = 0.331). The 3-year disease-free survival (DFS) rate was 72.0% in the TaTME group and 56.6% in the LapTME group (*p* = 0.038). In conclusion, better DFS and fewer LR events were observed after TaTME; thus, TaTME can be considered a safe and feasible approach in patients with low rectal cancer.

## 1. Introduction

According to GLOBOCAN statistics, there were an estimated 732 thousand newly diagnosed rectal cancer cases in 2020, and 339 thousand patients died [1]. To treat locally advanced rectal cancer with a multimodal approach, incorporating neoadjuvant (chemo) radiotherapy, surgery, and adjuvant chemotherapy has led to improved survival and reduced local recurrence [2]. Despite the trend of intensification of neoadjuvant therapy, surgery remains the mainstay treatment for rectal cancer. After the concept of total mesorectal excision (TME) was introduced by Dr. Heald in the 1980s, the local recurrence (LR) rate of rectal cancer decreased, and survival increased to 75–80% [3,4]. Good specimen quality with an adequate circumferential resection margin (CRM), which represents a good TME, is important and is associated with the oncological outcomes [5,6]. Currently, laparoscopic surgery is the standard treatment for rectal cancer, which offers better short-term outcomes by decreasing postoperative pain, has a better recovery, leads to a shorter hospital stay [7], and has comparable oncological outcomes to open surgery [8,9,10]. However, laparoscopic TME (LapTME) is more difficult in patients with obesity, narrow pelvis, male sex, bulky tumors, and low-lying tumors, and may result in suboptimal specimen quality [11]. To solve this problem, a transanal approach with “bottom–up” procedures was proposed. Sylla et al. first reported a successful case of transanal TME (TaTME) in 2010 using natural orifice transluminal laparoscopic surgery for rectosigmoid resection. TaTME is a safe procedure with acceptable perioperative and pathologic results [12].

Recently, TaTME has become a popular procedure for low- and mid-rectal cancers. Previous studies have showed that the transanal approach is an oncologically safe procedure and is effective for distal mesorectal dissection [13,14]. Compared with LapTME, TaTME provided less CRM involvement, less leakage, lower postoperative morbidity, and lower readmission rates. Moreover, the LR rate was comparable between LapTME and TaTME [14]. Regarding long-term oncological outcomes, a recent meta-analysis showed comparable LR rate, distant metastasis rate, 2-year overall survival, and 2-year disease-free survival between LapTME and TaTME, in which the functional outcomes and quality of life were also similar [15]. Despite the above benefits, diverse results have been obtained after the TaTME procedure. In Norwegian national data, an unexpected higher LR rate (11.6% at 2.4 years, compared to the national average of 2.4%) and multifocal growth patterns were reported [16,17]. To avoid unsatisfactory results, adequate TaTME training, proper case selection, and maintenance of high procedural volumes were recommended [18].

Currently, published observational studies usually compare TaTMEs with LapTMEs in low- to mid-rectal cancers. Ongoing randomized control trials (RCTs), the COLOR III trial, and the TaLAR trial also enrolled patients with rectal tumors ≤10 cm from the anal verge and below the peritoneal reflection, respectively [19,20]. We still need several years to observe the results of these RCTs. In our institution, we began performing the TaTME procedure in 2015, mostly for lower rectal cancers, because of the benefit of completeness of mesorectal excision and direct visualization of the distal resection margin. In this propensity score-matching (PSM) study, we aimed to analyze the early experience of TaTME surgery compared with LapTME surgery in patients with lower rectal cancer in our center. The primary endpoints were histopathological and perioperative outcomes. The secondary endpoints were overall survival (OS), disease-free survival (DFS), and local recurrence (LR) rate.

## 2. Materials and Methods

### 2.1. Patient Selection

This study was approved by the Institutional Review Board of Chang Gung Memorial Hospital (approval number 202200885B0). Informed consent was waived owing to the retrospective nature of the study. Data of consecutive patients diagnosed with lower rectal cancer whom underwent elective proctectomy at the Chang Gung Memorial Hospital between January 2015 and December 2019 was collected retrospectively. The patients fulfilling the inclusion criteria were enrolled: (1) pathologically proven rectal adenocarcinoma before surgery; (2) tumor located ≤6 cm from the anal verge, measured by rigid sigmoidoscopy; and (3) underwent laparoscopic restorative surgery, either laparoscopic TME or TaTME. Patients who did not fulfill the inclusion criteria were excluded from the analysis. The case selection flow chart is shown in Figure 1.

### 2.2. Assessment and Treatment Protocol

Before initiating therapy, all patients underwent complete staging, including physical examination, digital rectal examination, and colonoscopy, with or without rectal endoscopic ultrasonography. A chest, abdominal, and pelvic computed tomography (CT) survey was used to assess distant organ metastases. Pelvic magnetic resonance imaging (MRI) was performed to evaluate regional tumor conditions. If patients underwent neoadjuvant treatment, a re-staging workup usually involves physical examination, colonoscopy, and CT or MRI before surgery. Staging was conducted according to the 8th edition of the Union for International Cancer Control TNM classification. The treatment protocol for all patients was discussed at a multidisciplinary team meeting (MDT). Surgery, either LapTME or TaTME, was determined based on the surgeon’s preference. A protective colostomy/ileostomy was performed based on the surgeon’s judgment.

### 2.3. Data Collection

We retrieved and analyzed data from the Colorectal Section Tumor Registry at Chang Gung Memorial Hospital, which is a prospectively designed database consisting of records of postoperative patients who were consecutively and actively followed up. All data were recorded in the hospital database and were used for research purposes only.

Preoperative variables, including age, sex, body mass index (BMI), preoperative serum albumin level, and carcinoembryonic antigen (CEA) level were analyzed. Blood samples were obtained 1 week prior to surgery. Clinicopathological parameters, including neoadjuvant treatment methods, anastomosis methods (handsewn or staple), stoma creation, amount of blood loss, conversion to laparotomy, tumor size, tumor location, T stage, N stage, histological grade, histological type, distal resection margin (DRM), and CRM were also analyzed. Tumor stage was classified according to the eighth edition of the Union for International Cancer Control tumor-node-metastasis (TNM) classification. Postoperative outcomes, including postoperative complications, mortality, OS, DFS, LR, and distant metastasis (DM) rates, were assessed. Postoperative complications were classified according to the Clavien–Dindo (CD) grade.

### 2.4. Follow-Up

All physicians in this institution adopted the same follow-up routines and adjuvant treatment protocols. The follow-up program included physical examinations and CEA tests in clinic every 3–6 months for 3 years after the primary tumor resection. Chest radiography, abdominal ultrasonography, or abdominal CT imaging, in addition to colonoscopy, was performed one year postoperatively and then every 1 to 2 years when necessary. The follow-up status was recorded every 12 months in the Colorectal Section Tumor Registry. The date of first recurrence was defined as the first date when the existence of LR and/or distant metastases was confirmed by histology of biopsy specimens, additional surgery, and/or radiological studies. OS was defined as the interval from cancer resection to death or last follow-up. DFS was defined as the interval from cancer resection to the date of the first recurrence, death, or the last follow-up. The last follow-up date in this study was 31 May 2022.

### 2.5. Statistical Analysis

Data are presented as mean ± standard deviation or total number (%). The categorical variables were compared by Pearson’s chi-square test or Fisher’s exact test. The continuous variables were compared by Mann–Whitney U test. The survival analysis was performed using Kaplan–Meier curves and compared by the log-rank test. The univariate analysis was performed using the Cox proportional hazards model to assess risk factors associated with DFS. PSM was performed using a logistic regression model, with the operative method (LapTME vs. TaTME) set as the dependent variable. Patients were matched 1:1 using the neighbor-matching method. Patients in both groups were matched according to age, sex, BMI, American Society of Anesthesiology (ASA) score, post-treatment tumor size, distance from the anal verge, pathological T stage, pathological N stage, clinical M stage, and whether they received neoadjuvant treatment. Differences were considered statistically significant when a two-sided *p* < 0.05. All parameters were analyzed using the Statistical Package for Social Sciences (SPSS) version 24 (IBM Corp., New York, NY, USA) and GraphPad Prism version 9 (GraphPad Software Inc., San Diego, CA, USA).

## 3. Results

### 3.1. Patient Characteristics before and after Matching

A total of 216 patients were included in this study; of them, 118 underwent LapTME and 98 underwent TaTME surgery. In the unmatched analysis, patients who underwent TaTME were younger (58.70 ± 10.90 vs. 62.47 ± 13.03 years, *p* = 0.005), predominantly male (78.6% vs. 53.4%, *p* < 0.001), had higher BMI (mean 25.02 ± 3.83 vs. 23.67 ± 3.23, *p* = 0.01), lower rectal tumors from the anal verge (mean 4.21 ± 1.19 vs. 5.25 ± 0.92 cm, *p* < 0.001), and underwent more neoadjuvant treatment (65.3% vs. 39.8%, *p* < 0.001). The ASA classification, CEA level, albumin level, and pathological stage did not differ according to the patients’ baseline characteristics. After PSM, the study population included 63 patients in each group, and there were no significant differences between the characteristics of the LapTME and TaTME groups. The basic characteristics of the patients are shown in Table 1.

### 3.2. Operative Parameters and Short-Term Outcomes

The operative parameters are listed in Table 2. The mean operative time was longer in the TaTME group (394.29 ± 110.32 vs. 332.65 ± 101.13 min, *p* < 0.001). More patients in the TaTME group underwent handsewn anastomosis (49.2% vs. 3.2%, *p* < 0.001) and natural orifice specimen extraction procedures (79.4% vs. 15.9%, *p* < 0.001). There were no differences in the amount of blood loss, rate of conversion to laparotomy, and rate of diverting stoma between the LapTME and TaTME groups.

Postoperatively, 16 patients in the LapTME group and 11 in the TaTME group experienced surgical complications. Of them, six (9.5%) and four (6.3%) had CD classification ≥3 complications in the LapTME and TaTME groups, respectively (*p* = 0.098). Anastomosis leakage did not differ between the two groups (LapTME vs. TaTME:11.1% vs. 7.9%, *p* = 0.544). Of these, four and three patients underwent reoperation for diverting stoma creation, respectively. There were no differences in prolonged ileus and intra-abdominal infections between the LapTME and TaTME groups. No postoperative mortality occurred in any of the groups.

The mean days until the removal of urinary catheters were 5.63 days in the LapTME group and 6.21 days in TaTME group (*p* = 0.211). There was no difference in the time to first flatus passage, first stool passage, tolerated liquid diet, and tolerated soft diet between the two groups. The mean length of hospital stay was 10.21 ± 6.39 days in LapTME group and 10.71 ± 4.97 days in TaTME group, respectively (*p* = 0.062). Postoperative complications and short-term outcomes are listed in Table 3.

### 3.3. Histopathological Parameters

The pathological results are presented in Table 4. Five (7.9%) patients in the LapTME group and seven (11.1%) in the TaTME group had a pathological complete response (*p* = 0.544). Positive CRM was found in seven (11.1%) and three (4.8%) patients in the LapTME and TaTME groups, respectively (*p* = 0.187). No difference in mean DRM was found between two groups (LapTME vs. TaTME:1.12 ± 1.05 cm vs. 1.22 ± 0.84 cm, *p* = 0.19). None of the patients in the TaTME group had a positive distal margin, except for two patients in the LapTME group (*p* = 0.496). Overall, R1 resection was found in eight (12.7%) and three (4.8%) patients in the LapTME and TaTME groups, respectively (*p* = 0.205). There were no significant differences in histological type, lymph node number, lymphovascular invasion, or perineural invasion. However, seven (11.1%) patients in the LapTME group had grade III adenocarcinoma, in contrast to three (4.8%) patients in the TaTME group (*p* = 0.035).

### 3.4. Long-Term Outcomes

The mean follow-up time was 41.33 and 47.7 months in the LapTME and TaTME groups, respectively (Table 5). LR occurred in 15 (23.8%) patients in the LapTME group and six (9.5%) patients in the TaTME group (*p* = 0.031). Of the 15 patients with LR in the LapTME group, seven were found in the local region, seven in the pelvic lymph nodes, and one in the presacral region. Five patients in the TaTME group and one in the TaTME group had LR in the local region and pelvic side wall, respectively. The median time to LR was 10.1 and 21.2 months in LapTME group and TaTME group, respectively (*p* = 0.276). After the end of follow-up in this study, 19 (30.2%) and 15 (23.8%) patients were deceased in the LapTME and TaTME groups, respectively. During the study period, the permanent stoma rate was 24.2% in the LapTME group and 15.9% in the TaTME group (*p* = 0.245).

Regarding the oncological outcome, there was no difference in the 3-year OS in both groups (LapTME vs. TaTME:73.60% vs. 80.30%, *p* = 0.331) (Figure 2). However, there was a significant difference in the 3-year disease-free survival in patients with stage I–III disease (LapTME vs. TaTME:56.60% vs. 72.00%, *p* = 0.038) (Figure 3). The estimated LR rate at three years was 6.80% in the TaTME group and 22.50% in the LapTME group (*p* = 0.014) (Figure 4). The estimated DM rate at 3 years was 20.00% in the TaTME group and 30.80% in the LapTME group (*p* = 0.081) (Figure 5). The risk factors for 3-year DFS are summarized in Figure 6. R1 resection, CRM involvement, preoperative CEA level ≥ 5 ng/mL, perineural invasion, lymphovascular invasion, pathological N2 stage, and T4 stage were poor prognostic factors for DFS. Patients who underwent TaTME had a better 3-year DFS than those who underwent LapTME (HRs: 0.537; 95% CI: 0.296–0.975; *p* = 0.041).

## 4. Discussion

This study showed that for low rectal cancer, TaTME had comparable perioperative outcomes, postoperative recovery, complications, and histopathological outcomes with LapTME, although longer operative times were required. Regarding the oncological outcome, TaTME had similar 3-year OS and DM rates as LapTME, but superior 3-year DFS and LR rates were observed in our study.

Surgery for rectal cancer remains challenging, because the rectum is covered by fatty tissue and anatomically located in the narrow pelvic space, and good visualization of the mesorectal structure is difficult during surgery [3]. With advances in surgical instruments and techniques, laparoscopic TME has been gradually adopted as the standard approach after several large RCTs showed improved short-term outcomes and equivalent oncological results compared to open TME surgery. However, the reported CRM involvement rate was diverse, ranging from 2.9% to 12.1%, compared with 4.1–10.0% in the open surgery group [8,9,10,21,22]. For lower rectal cancer, it is more difficult to perform an adequate mesorectal excision downward to the pelvic floor; thus, a transanal approach, which ensures a better vision of the mesorectal plane and exact DRM, has been proposed and fashioned in recent decades [23].

In our study, the CRM involvement rates were 4.8% and 11.1% in the TaTME and LapTME groups, respectively. No distal margin involvement was observed in the TaTME group. These results are consistent with those of previous studies. In a recent meta-analysis that compared TaTME with LapTME for low- and middle-rectal cancer, the pooled results from 14 observational studies showed a CRM involvement rate of 4% in TaTME and 8.8% in LapTME (OR 0.48, 95% CI 0.27–0.86; *p* = 0.01). Distal margin involvement in five studies was 1.4% in each group, and no difference was observed [14]. As an independent poor prognostic factor for LR and survival, CRM involvement is an important surrogate for rectal cancer surgery [6]. Herein, we included patients with lower rectal cancer, which represents a subgroup requiring the most difficult surgical approach, and the histopathological results in the TaTME group were similar to those in the LapTME group. A study conducted by Roodbeen et al. comparing TaTME with LapTME in MRI-defined low rectal cancer also showed a similar CRM involvement between the two approaches (TaTME vs. LapTME: 4.9% vs. 12.2%, *p* = 0.432) [24]. In addition to surgical margin involvement, no differences in the R1 resection rate and harvested lymph node numbers were observed in our study. These findings showed that TaTME is a feasible approach for patients with low rectal cancer.

In the present study, no differences in blood loss, conversion rate, or postoperative complications were observed between the TaTME and LapTME group. A 6.3% rate of major complications was observed in the TaTME group in this study, which is equivalent to the international registry of TaTME patients, which showed a 10.9% CD classification III–IV complications [13]. The majority of postoperative complications in this study were anastomosis leakage, which occurred in 7.9% of the TaTME group and 11.1% of the LapTME group, which is similar to the pooled results from a previous meta-analysis (leakage rate: TaTME vs. LapTME:6.4 vs. 11.6%, OR 0.53, 95%CI 0.31–0.93; *p* = 0.03) [14]. Urethral injury, which is considered a TaTME-specific complication, did not occur in our study. The length of hospital stay and recovery of bowel function were also similar between the TaTME and LapTME groups. Although with equivalent short-term outcomes, a significantly longer operative time was observed in the TaTME group in our study, with an average of 60 min longer than that in the LapTME group. This may be attributed to the learning curve of the TaTME procedure. Previous studies estimated a learning curve for TaTME in 40–50 cases [25,26]. A two-team approach can also decrease the operative time and conversion rate [26]. In our hospital, surgeons have used the TaTME procedure for low rectal cancers since 2015, and this study included the early experience of this transanal approach. Nevertheless, the good short-term outcomes in the present study indicate that TaTME can be safely applied during the learning curve of surgeons experienced in LapTME.

Currently, there are few studies that provide long-term follow-up data. The 2- to 3-year OS of TaTME was reported to be between 93.3% and 96.0% [24,27,28,29,30], and these results are similar to those of LapTME. With a mean follow-up time of over 41 months, the 3-year OS in this study showed no significant difference between TaTME and LapTME, which is consistent with previous reports. Unlike other reports, which had similar DFS between TaTME and LapTME, a significantly worse 3-year DFS in the LapTME group was observed in our study. With the exclusion of initial stage IV disease, our LapTME cohort had a 3-year DFS rate of 56.6%, in contrast to 72.0% in the TaTME cohort. The main reason for the worse DFS was the high incidence of LR in the LapTME group. The 3-year LR rates were 22.5% and 6.8% in the LapTME and TaTME groups, respectively. Although there were no statistically significant differences in CRM and DRM involvement between the TaTME and LapTME groups in our study. The R1 resection rate was higher in the LapTME group, accounting for 12.7%, in contrast to 4.8% in the TaTME group. In the post-hoc analysis, six of the eight patients who underwent R1 resection had local recurrence after surgery in the LapTME group; however, one of the three patients in the TaTME group had LR. Another possible reason for the high LR rate may be the low implementation rate of neoadjuvant treatment in the LapTME group, with 47.6% of patients receiving surgery directly, although no statistically significant difference was observed compared with the counterpart of the study cohort. However, because no uniform therapy was applied in the neoadjuvant treatment, multiple neoadjuvant treatment groups for the two cohorts makes it hard to interpret the true effect of neoadjuvant treatment on DFS. The third reason may result in a poor DFS in the LapTME group is the worse histological grade than the TaTME group. Moreover, although without significant difference after PSM, there still more T4 and N1/2 tumors in the LapTME group as opposed to the TaTMA group. The T4 and node positive tumors will inherently have a higher LR rate. However, our results imply that for low rectal cancer, improved CRM quality by TaTME may have a positive effect on long-term oncological outcomes.

The reported LR rate from current studies ranged from 2.6% to 3.8%, and the DFS ranged from 78.8% to 86%, with a median follow-up time of 2–3 years [24,27,28,29,30]. These results were all from observational studies and may be affected by the experience and learning curve of surgeons. Recently, Larsen et al. reported an LR rate of 9.5% in 110 patients who underwent TaTME with a median of 11 months follow-up in Norway. A rapid and unusually multifocal growth pattern in the pelvic cavity and sidewall has been reported, which differs from conventional TME surgery. Thus, the moratorium for TaTME in Norway was announced [16]. In contrast, one analysis of 159 consecutive patients who underwent TaTME at two high-volume referral centers in the Netherlands, with a minimum follow-up of 36 months, showed a 5-year LR rate of 4.0% and 5-year DFS of 81%. The authors indicated that TaTME is a safe and feasible approach for high-volume hospitals [31]. Currently, an emphasis has been placed on structured training programs before TaTME, proper selection of cases, and maintenance of high procedure volumes for optimal outcomes [18].

A previous study revealed that the level of rectal cancer significantly affects the oncological outcomes. Chiang et al. reported a 3-year DFS of 47.39% in patients with low rectal cancer compared to 67.31% in patients with mid-rectal cancer who underwent curative-intent surgery without neoadjuvant therapy [32]. In the present study, we only included patients with low rectal cancer who underwent restorative rectal surgery to assess the safety and outcome of the TaTME procedure because surgeons in our institute began adopting this approach mainly for low rectal cancer because of its good TME quality compared with the conventional laparoscopic approach. With similar histopathological and perioperative outcomes and an improved 3-year DFS observed in the present study, we believe that TaTME is a feasible approach for patients with low rectal cancer. With the cumulative experience of surgeons, the utility of TaTME can be further expanded to more complex situations such as reoperative cases, local recurrences, and favorable stage IV disease [33].

Our study has some limitations that need to be considered. First, data regarding the pathology reports of TME grades to assess the quality of TME specimens is lacking. Second, there were no functional differences between TaTME and LapTME. Third, this study was limited by its retrospective nature, and the characteristics of the participants and the selection of treatment may be biased; however, by applying PSM, the demographic characteristics of both groups used in the analysis were the same. Fourth, the sample size was small in this study, especially after the PSM, and the statistical power may be lower than the larger trials. Finally, the study represented the initial experience of TaTME in one center, and surgeons are still under the learning curve of the procedure. Our results may not demonstrate the precise advantages of the TaTME. Nonetheless, the fact that TaTME provided equivalent short-term outcomes, had compatible OS, and a better DFS compared to LapTME in this study still implies the feasibility and oncological safety of using TaTME for low rectal cancer.

## 5. Conclusions

TaTME is a more demanding technique with a longer learning curve than that of LapTME. Our experience revealed that TaTME had similar histopathological results and postoperative outcomes as LapTME, even in the learning curve. Better CRM and DRM observed in TaTME may contribute to better DFS and fewer LR events. TaTME can be considered a safe and feasible approach in patients with low rectal cancer.

## Figures and Tables

**Figure 1 cancers-14-04098-f001:**
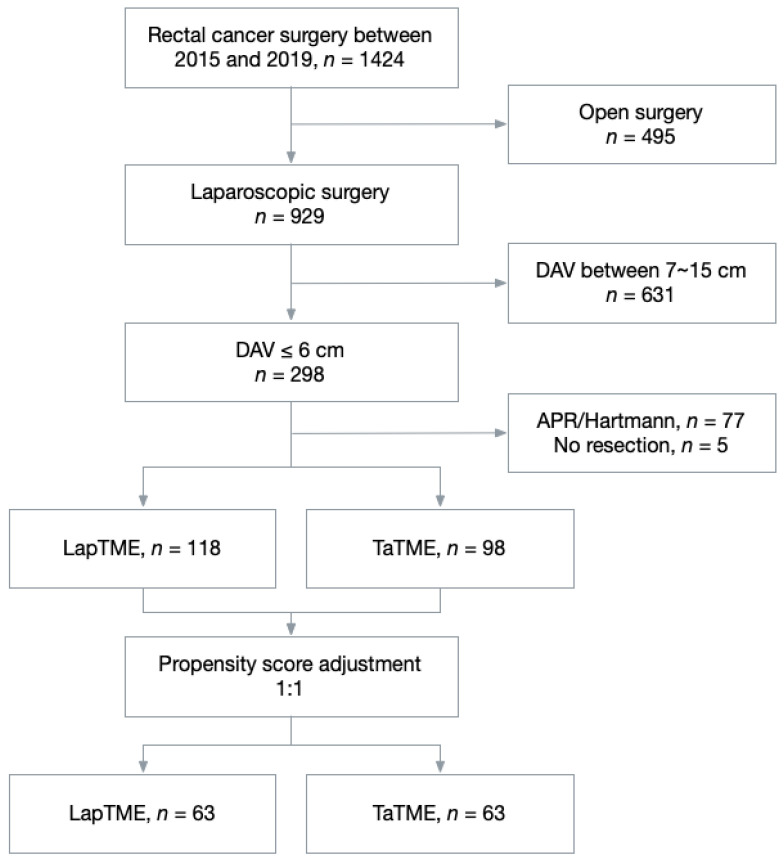
The flowchart illustrating the clinical patient selection in this study.

**Figure 2 cancers-14-04098-f002:**
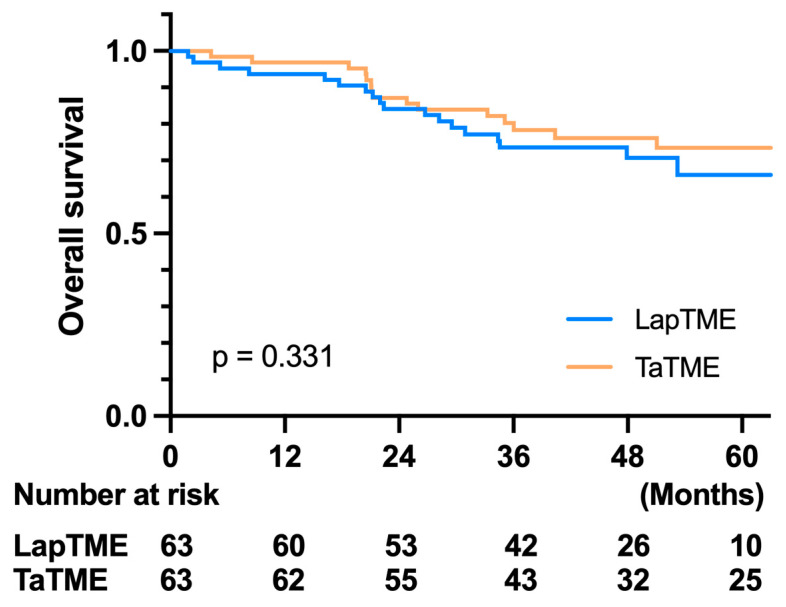
Kaplan–Meier survival curves of OS between the TaTME and LapTME group.

**Figure 3 cancers-14-04098-f003:**
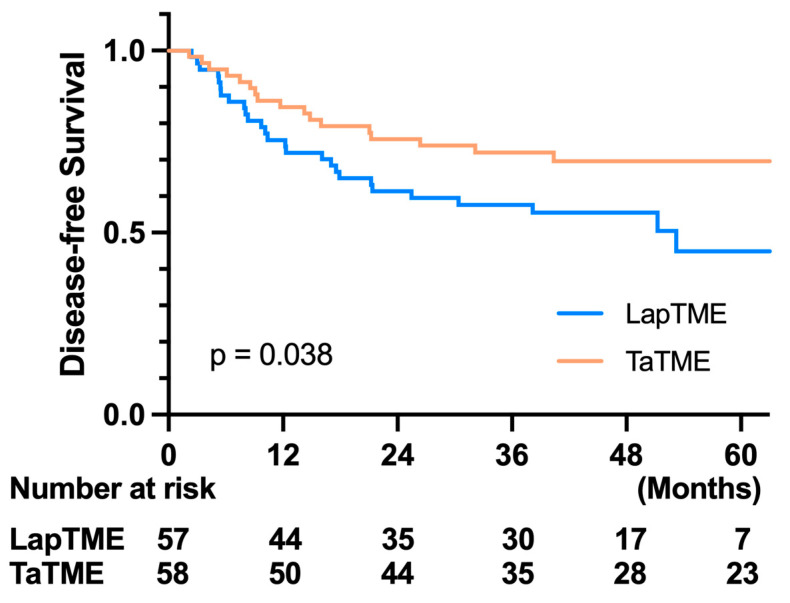
Kaplan–Meier survival curves of DFS between the TaTME and LapTME groups.

**Figure 4 cancers-14-04098-f004:**
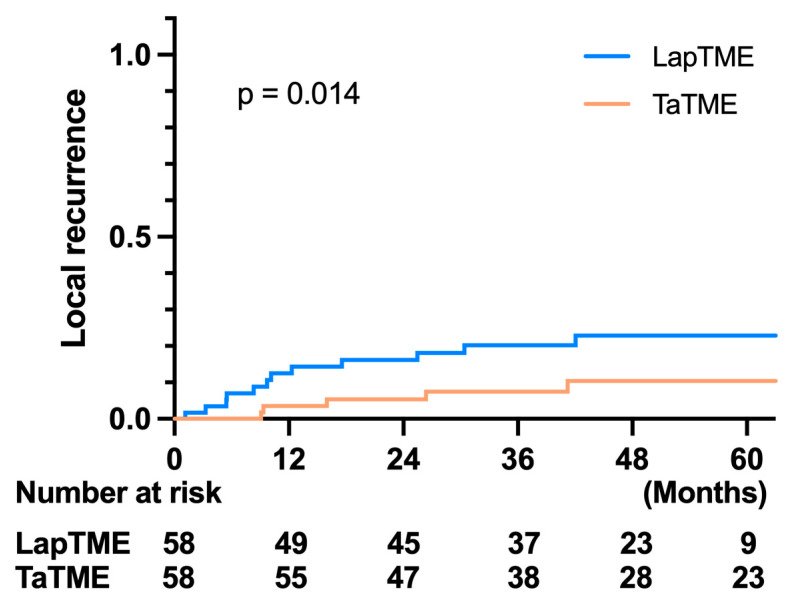
Kaplan–Meier survival curves of LR between the TaTME and LapTME groups.

**Figure 5 cancers-14-04098-f005:**
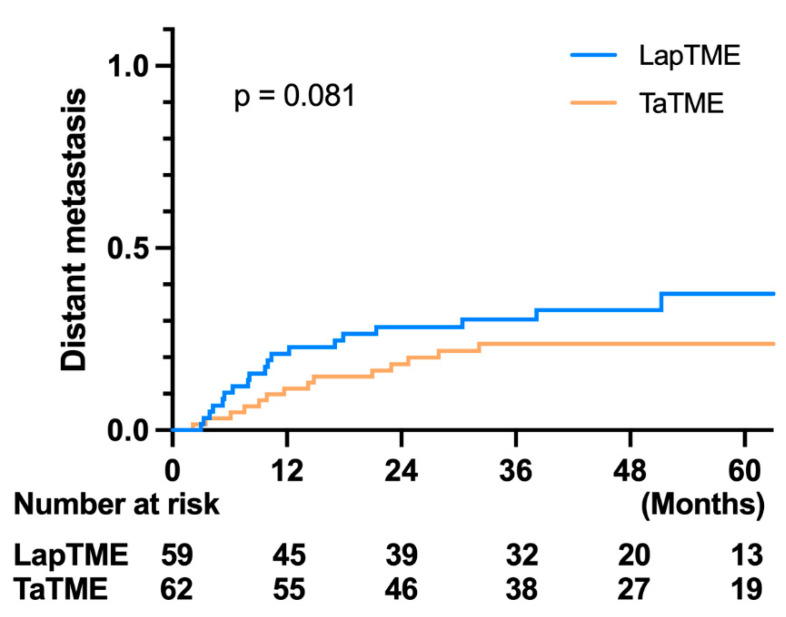
Kaplan–Meier survival curves of DM between the TaTME and LapTME groups.

**Figure 6 cancers-14-04098-f006:**
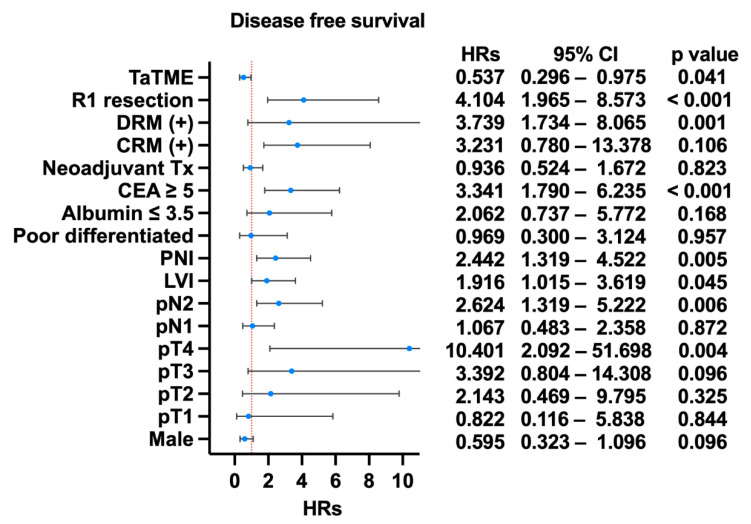
Summary of the risk factors for 3-year DFS.

**Table 1 cancers-14-04098-t001:** Basic characteristics of patients with low rectal cancer who underwent restorative proctectomy before and after propensity score matching.

	Before Propensity Score Matching			After Propensity Score Matching		
	LapTME (*n* = 118)	TaTME (*n* = 98)	*p* Value	LapTME (*n* = 63)	TaTME (*n* = 63)	*p* Value
Age	62.47 ± 13.03	58.70 ± 10.90	0.005	62.10 ± 12.37	60.35 ± 11.82	0.331
Gender						
Male	63 (53.4)	77 (78.6)	<0.001	41 (65.1)	48 (76.2)	0.171
Female	55 (46.6)	21 (21.4)		22 (34.9)	15 (23.8)	
BMI	23.67 ± 3.23	25.02 ± 3.83	0.01	23.91 ± 3.20	24.63 ± 4.09	0.326
Distance from AV	5.25 ± 0.92	4.21 ± 1.19	<0.001	5.00 ± 1.02	4.76 ± 0.93	0.084
Neoadjuvant treatment						
Yes	47 (39.8)	64 (65.3)	<0.001	33 (52.4)	38 (60.3)	0.369
No	71 (60.2)	34 (34.7)		30 (47.6)	25 (39.7)	
Albumin						
<3.5	5 (4.2)	4 (4.2)	1	2 (3.2)	4 (6.6)	0.38
≥3.5	113 (85.8)	92 (95.8)		61 (96.8)	57 (93.4)	
CEA						
<5	94 (79.7)	83 (84.7)	0.338	46 (73.0)	54 (85.7)	0.078
≥5	24 (20.3)	15 (15.3)		17 (27.0)	9 (14.3)	
ASA score						
2	51 (43.2)	41 (41.8)	0.838	19 (30.2)	23 (36.5)	0.45
3	67 (56.8)	57 (58.2)		44 (69.8)	40 (63.5)	
pT-stage						
T0	9 (7.6)	9 (9.2)	0.695	5 (7.9)	7 (11.1)	0.494
T1	14 (11.9)	12 (12.2)		7 (11.1)	8 (12.7)	
T2	36 (30.5)	28 (28.6)		17 (27.0)	14 (22.2)	
T3	52 (44.1)	47 (48.0)		29 (46.0)	33 (52.4)	
T4	7 (5.9)	2 (2.0)		5 (7.9)	1 (1.6)	
pN-stage						
N0	85 (72.0)	64 (65.3)	0.307	42 (66.7)	41 (65.1)	0.74
N1	17 (14.4)	22 (22.4)		10 (15.9)	13 (20.6)	
N2	16 (13.6)	12 (12.2)		11 (17.5)	9 (14.3)	
M-stage						
M0	113 (95.8)	88 (89.8)	0.086	58 (92.1)	58 (92.1)	1
M1	5 (4.2)	10 (10.2)		5 (7.9)	5 (7.9)	
Neoadjuvant treatment						
No	71 (60.2)	34 (34.7)	<0.001	30 (47.6%)	25 (39.7)	0.506
Conventional CRT	18 (15.3)	37 (37.8)		16 (25.4)	22 (34.9)	
SCRT	11 (9.3)	9 (9.2)		5 (7.9)	7 (11.1)	
SCRT + CCT	15 (12.7)	18 (18.4)		10 (15.9)	9 (14.3)	
Chemotherapy	3 (2.5)	0		2 (3.2)	0	

LapTME: laparoscopic total mesorectal excision; TaTME: transanal total mesorectal excision; BMI: body mass index; AV: anal verge; ASA score: American Society of Anesthesiology score; CRT: chemoradiotherapy; SCRT: short course radiotherapy; CCT: consolidation chemotherapy; CEA: Carcinoembryonic Antigen.

**Table 2 cancers-14-04098-t002:** Post-matching of operative parameters among patients with low rectal cancer underwent restorative proctectomy.

	LapTME (*n* = 63)	TaTME (*n* = 63)	*p* Value
Operative time	332.65 ± 101.13	394.29 ± 110.32	<0.001
Blood loss			
<100 mL	49 (77.8)	41 (65.1)	0.115
≥100 mL	14 (22.2)	22 (34.9)	
Diverting stoma			
yes	47 (74.6)	54 (85.7)	0.118
no	16 (25.4)	9 (14.3)	
Conversion			
yes	2 (3.2)	0	0.496
no	61 (96.8)	63 (100)	
Anastomosis methods			
Hand sewn	2 (3.2)	31 (49.2)	<0.001
Staples	61 (96.8)	32 (50.8)	
Specimen extraction methods			
Right/Left lower incision	51 (81.0)	12 (19)	<0.001
NOSE	10 (15.9)	50 (79.4)	
Pfannenstiel incision	2 (3.2)	1 (1.6)	

LapTME: laparoscopic total mesorectal excision; TaTME: transanal total mesorectal excision; NOSE: natural orifice specimen retraction.

**Table 3 cancers-14-04098-t003:** Post-matching of short-term outcomes among patients with low rectal cancer underwent restorative proctectomy.

	LapTME (*n* = 63)	TaTME (*n* = 63)	*p* Value
Hospital stay	10.21 ± 6.39	10.71 ± 4.97	0.062
First flatus passage	2.32 ± 2.12	2.29 ± 1.50	0.381
First stool passage	3.38 ± 2.63	3.21 ± 2.49	0.872
Tolerated liquid diet	4.13 ± 3.43	4.05 ± 3.58	0.82
Tolerated soft diet	5.97 ± 4.55	5.75 ± 4.04	0.793
Remove Foley day	5.63 ± 3.15	6.21 ± 3.84	0.211
Clavien–Dindo Classification			
I	5 (7.9)	0	0.098
II	5 (7.9)	7 (11.1)	
III	4 (6.3)	4 (6.3)	
IV	2 (3.2)	0	
V	0	0	
Complication type			
Ileus	3 (4.8)	2 (3.2)	1
Anasomosis leak	7 (11.1)	5 (7.9)	0.544
IAI	7 (11.1)	6 (9.5)	0.77
Others	5 (7.9)	2 (3.2)	0.44
Re-operation			
Leakage	4 (6.3)	3 (4.8)	1
Bowel obstruction	1 (1.6)	1 (1.6)	

LapTME: laparoscopic total mesorectal excision; TaTME: transanal total mesorectal excision; IAI: intra-abdominal infection.

**Table 4 cancers-14-04098-t004:** Post-matching pathological finding among patients with low rectal cancer underwent restorative proctectomy.

	LapTME (*n* = 63)	TaTME (*n* = 63)	*p* Value
pCR	5 (7.9)	7 (11.1)	0.544
Histology type			
Adenocarcinoma	59 (93.7)	59 (93.7)	1
Signet ring cell/Mucinous	4 (6.3)	3 (4.8)	
Other	0	1 (1.5)	
Histology Grade			
Grade I/II	56 (88.9)	55 (87.3)	0.035
Grade III	7 (11.1)	3 (4.8)	
Unclassified	0	5 (7.9)	
Lymphovascular invasion			
positive	16 (25.8)	11 (18.0)	0.298
negative	46 (74.2)	50 (82.0)	
Perineural invasion			
positive	15 (24.2)	14 (23)	0.871
negative	47 (75.8)	7 (77)	
CRM			
Positive	7 (11.1)	3 (4.8)	0.187
Negative	56 (88.9)	60 (95.2)	
Distal resection margin, length	1.12 ± 1.05	1.22 ± 0.84	0.19
Distal resection margin			
positive	2 (3.2)	0 (0)	0.496
negative	61 (96.8)	63 (100)	
Lymph node yield	27.43 ± 11.66	25.44 ± 13.79	0.192
R1 resection	8 (12.7)	3 (4.8)	0.205

LapTME: laparoscopic total mesorectal excision; TaTME: transanal total mesorectal excision; pCR: pathological complete response; CRM: circumferential resection margin.

**Table 5 cancers-14-04098-t005:** Post-matching of long-term outcomes among patients with low rectal cancer underwent restorative proctectomy.

	LapTME (*n* = 63)	TaTME (*n* = 63)	*p* Value
Mean follow up time (months)	41.33 ± 17.72	47.7 ± 20.69	0.108
Local recurrence	15 (23.8)	6 (9.5)	0.031
Median Time to LR (months)	10.1 (1.2–63.4)	21.2 (9.1–67.8)	0.276
Distant metasasis *	19 (32.8)	12 (20.7)	0.142
Median Time to DM (months)	8.1 (1.0–51.3)	12.9 (2.2–76.9)	0.351
Deceased	19 (30.2)	15 (23.8)	0.422
3-year LR rate	22.50%	6.80%	0.014
3-year DM rate *	30.80%	20.00%	0.081
3-year overall survival	73.60%	80.30%	0.331
3-year disease free survival *	56.60%	72.00%	0.038
Permanent stoma	13 (21.0%)	9 (14.3%)	0.327

* Exclude 5 stage IV cases in each group; LR: local recurrence; DM: distant metastasis; LapTME: laparoscopic total mesorectal excision; TaTME: transanal total mesorectal excision.

## Data Availability

The datasets generated and analyzed during the current study are available from the corresponding author upon reasonable request.

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
