# Peer review of "Transanal Total Mesorectal Excision (TaTME) versus Laparoscopic Total Mesorectal Excision for Lower Rectal Cancer: A Propensity Score-Matched Analysis"

_cancers, 2022, doi:10.3390/cancers14174098_

Round 1
Reviewer 1 Report
This a single institution retrospective review of rectal cancer patients that underwent either laparoscopic LAR versus TaTME.
1. Was distance from the anal verge defined by examination (DRE), scope or MRI? Please define DAV in the paper.
2. Why was 6 cm chosen instead of 10cm?
3. For PSM was tumor size and DAV based on pretreatment or post treatment?
4. I don’t understand the ileus group in the re-operative section of table 3.
5. It looks like the lap group had worse histological grade than the TaTME group. With no difference in CrM and DRM between the two this could be one of the reasons for differences in DFS.
6. The multiple neoadjuvant treatment groups for the two cohorts makes it hard to interpret if it really the differences in surgical technique, even after PSM, versus the neoadjuvant treatment in terms of outcome as there are so few patients in each individual group (max thirty patients).
Reviewer 2 Report
This looks to be a good study and interesting outcomes. The only thing I noted was that despite propensity matching, there still seemed to be more T4 and N1/2 tumors in the Lap TME arm as opposed to the TaTMA arm. T4 and node positive tumors will inherently have a higher LR rate.
Also, the sample size is small with just 63 patients in each arm. I would be interested in seeing and update with larger numbers. This is also a limitation of your study.
